# One Pass Streaming Algorithm for Super Long Token Attention Approximation in Sublinear Space

## Abstract

Attention computation takes both the time complexity of $O(n^2)$ and the space complexity of $O(n^2)$ simultaneously, which makes deploying Large Language Models (LLMs) in streaming applications that involve long contexts requiring substantial computational resources. In recent OpenAI DevDay (Nov 6, 2023), OpenAI released a new model that is able to support a 128K-long document, in our paper, we focus on the memory-efficient issue when context length $n$ is much greater than 128K ($n \gg 2^d$). Considering a single-layer self-attention with Query, Key, and Value matrices $Q, K, V \in \mathbb{R}^{n \times d}$, the polynomial method approximates the attention output $T \in \mathbb{R}^{n \times d}$. It accomplishes this by constructing $U_1, U_2 \in \mathbb{R}^{n \times t}$ to expedite attention $\mathsf{Attn}(Q, K, V)$ computation within $n^{1+o(1)}$ time executions. Despite this, computing the approximated attention matrix $U_1 U_2^\top \in \mathbb{R}^{n \times n}$ still necessitates $O(n^2)$ space, leading to significant memory usage. In response to these challenges, we introduce a new algorithm that only reads one pass of the data in a streaming fashion. This method employs sublinear space $o(n)$ to store three sketch matrices, alleviating the need for exact $K, V$ storage. Notably, our algorithm exhibits exceptional memory-efficient performance with super-long tokens. As the token length $n$ increases, our error guarantee diminishes while the memory usage remains nearly constant. This unique attribute underscores the potential of our technique in efficiently handling LLMs in streaming applications.

## 1 Introduction

Large Language Models (LLMs) such as ChatGPT (ChatGPT, 2022), InstructGPT (Ouyang et al., 2022), Palm (Chowdhery et al., 2022; Anil et al., 2023), BARD (BARD, 2023), GPT-4 (OpenAI, 2023), LLAMA (Touvron et al., 2023a), LLAMA 2 (Touvron et al., 2023b), Adobe firefly (Adobe, 2023), have revolutionized various aspects of human work. These models have shown remarkable capabilities in dialog systems (Ni et al., 2023; Deng et al., 2023a;b), document summarization (Huang et al., 2023; Ghadimi & Beigy, 2023; Zhang et al., 2023; Krishna et al., 2023), code completion (Zheng et al., 2023; Liu et al., 2023a; Allal et al., 2023), and question-answering (Rogers et al., 2023; Budler et al., 2023; Roy et al., 2023). However, their performance in these applications is often constrained by the context length.

To prepare for the coming of artificial general intelligence (AGI) (Bubeck et al., 2023), one of the crucial bottlenecks for nowadays LLM is about how to handle super long context. In recent OpenAI DevDay (Nov 6, 2023) (Altman, 2023) [1], OpenAI released a new model that is able to support a 128K-long document. In other words, you can feed a 300-page textbook into LLM. This is already quite surprising. However, to finally achieve AGI, we might need to feed some data that is significantly larger than the memory in a model. For example, what if we can't even store the entire $x$ pages of a book in memory when $x$ is super large?

A longer context length allows the LLM to incorporate more information, potentially leading to more accurate and contextually appropriate responses. This increased capacity for information processing can enhance the LLM's understanding, coherence, and contextual reasoning abilities. Therefore, to optimally utilize pretrained LLMs, it's crucial to efficiently and accurately generate long sequences.

---

[1]OpenAI DevDay, Opening Keynote. https://www.youtube.com/watch?v=U9mJuUkhUzk

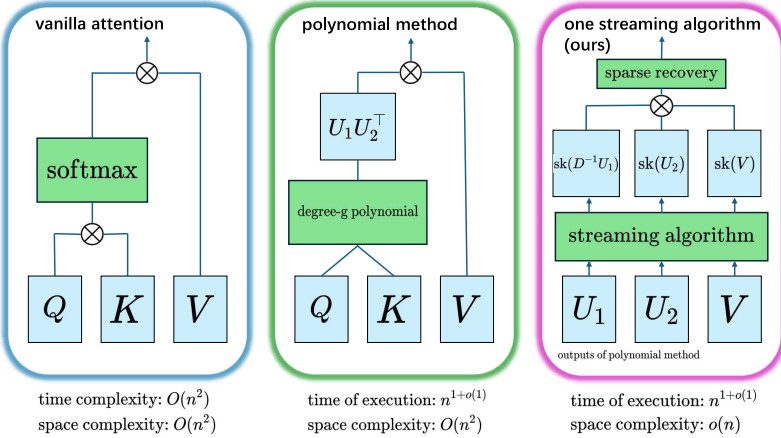

Figure 1: Comparison between our method and previous works. On the left: vanilla attention computation (Vaswani et al., 2017); Middle: fast attention by polynomial method (Alman & Song, 2023a); On the right: one pass algorithm (ours).

Despite the advantages of a long context length, LLMs, especially those based on transformers, face significant computational challenges. Inference with a long context in LLMs is computationally intensive, requiring both $O(n^2)$ space complexity and $O(n^2)$ time complexity to compute the attention output. This computational demand can limit the practical application of LLMs in real-world scenarios, making it a crucial area for further research and optimization.

Previous work (Alman & Song, 2023a;b) has conducted an in-depth study on the fast approximation of attention computation within $n^{1+o(1)}$ time executions without space requirements. Below is a formal definition:

**Definition 1.1** (Static Attention Approximation without Space Requirement (Alman & Song, 2023a))**.** *Let $\epsilon \in (0, 1)$ denote an accuracy parameter. Given three matrices $Q, K, V \in \mathbb{R}^{n \times d}$, the goal is to construct $T \in \mathbb{R}^{n \times d}$ such that*

$$\|T - \mathsf{Attn}(Q, K, V)\|_\infty \le \epsilon$$

*where*

- $\mathsf{Att}(Q, K, V) := D^{-1}AV$

- $A \in \mathbb{R}^{n \times n}$ *is a square matrix $A := \exp(QK^\top/d)$, here we apply $\exp()$ function entrywisely.*

- $D \in \mathbb{R}^{n \times n}$ *is a diagonal matrix $D := \mathrm{diag}(A\mathbf{1}_n)$ where $\mathbf{1}_n \in \mathbb{R}^n$ is a length-$n$ vector where all the entries are ones.*

However, the memory requirement of caching attention matrix $D^{-1}A$ for LLM's inference is still a considerable issue that consumes $O(n^2)$ space complexity. In this paper, we study the computation-efficiency problem in the context of transformer-based LLMs with super long context. We consider the following problem:

*How can we compute the attention with super-long context in space complexity of $o(n)$?*

This question is crucial as it directly relates to the computational efficiency of LLMs, particularly when dealing with super-long context lengths. In response to this question, our goal is to develop an effective streaming algorithm. We aim to define and solve the streaming version of approximate attention computation, which is a critical aspect of our research. By addressing this problem, we hope to significantly enhance the computational efficiency of transformer-based LLMs, thereby expanding their applicability in various real-world scenarios. We define the streaming version of approximate attention computation, which is also the problem we aim to solve in this paper:

**Definition 1.2** (Streaming Attention Approximation with Sublinear in $n$ Space). *Given $Q, K, V \in \mathbb{R}^{n \times d}$, we're only allowed to use $o(n)$ spaces and read $Q, K, V$ in one pass, and then outputs $T \in \mathbb{R}^{n \times d}$ such that $T$ is close to $D^{-1}AV$.*

## 1.1 OUR RESULT

In our research, we tackle the challenge of efficiently calculating attention for extremely long input sequences (super-long context) with limited memory resources. Our goal is to process Query $Q$, Key $K$, and Value $V$ matrices of size $n \times d$ in a single streaming pass while utilizing only $o(n)$ space.

To address this, we propose a novel one-pass streaming algorithm (Algorithm 1). For $d = O(\log n)$, we first compute low-rank approximation matrices $U_1, U_2 \in \mathbb{R}^{n \times t}$ as in prior work (Alman & Song, 2023a) such that $D^{-1}U_1U_2^\top \approx \mathsf{Attn}(Q, K, V)$.

Next, we introduce sketching matrices $\Phi \in \mathbb{R}^{m_1 \times n}$ (Nakos & Song, 2019), $\Psi \in \mathbb{R}^{m_2 \times n}$ (Alon et al., 1999) to sample $U_1, U_2$ respectively, where $m_1 = O(\epsilon_1^{-1}k \log n), m_2 = \mathcal{O}(\epsilon_2^{-2} \log n)$ and $k$ controls sparsity.

We present our main result as follows:

**Theorem 1.3** (Main Result, informal version of Theorem 4.1). *There is a one pass streaming algorithm (Algorithm 1) that reads $Q, K, V \in \mathbb{R}^{n \times d}$ with $d = O(\log n)$, uses $O(\epsilon_1^{-1}kn^{o(1)} + \epsilon_2^{-2}n^{o(1)})$ spaces and outputs a matrix $T \in \mathbb{R}^{n \times d}$ such that*

- *For each $i \in [d]$, $T_{*,i} \in \mathbb{R}^n$ is $O(k)$-sparse column vector*

- *For each $i \in [d]$, $\|T_i - y_i\|_2 \leq (1+\epsilon) \cdot \min_{k-\text{sparse } y'} \|y' - y_i\|_2 + \epsilon_2$ where $y_i = D^{-1}AV_{*,i}$*

- *The succeed probability $0.99$.*

The purpose of our work is to address the memory constraints associated with computing attention over very long sequences where the context length $n \gg 2^d$ (potentially infinitely long), then furthermore contribute towards more efficient and scalable transformer models, which could assist in advancing capabilities towards artificial general intelligence (AGI) (Bubeck et al., 2023). Section 2 discusses related work that focuses on approximating attention computation. This includes prior studies on fast approximations without space requirements, which lay the groundwork for our streaming formulation. In Section 3, we outline the preliminary concepts and definitions used in our analysis. This includes problem definition, attention computation, and sketching techniques. Our key technical contributions are presented in Section 4. Here, we introduce a novel one-pass streaming algorithm for attention approximation with sublinear $o(n)$ space complexity. We also state our main theorem, which establishes performance guarantees for our proposed algorithm. Section 4 further provides a detailed proof of the main theorem. This validates that our algorithm is able to process queries, keys and values in a single streaming pass while meeting the stated approximation bounds using limited memory.

## 2 RELATED WORK

In this section, we briefly review three topics that have close connections to this paper, which are Attention Theory, Streaming Algorithm and Improving LLM's Utilization of Long Text.

**Attention Theory** Numerous recent studies have explored attention computation in Large Language Models (LLMs) (Kitaev et al., 2020; Tay et al., 2020; Chen et al., 2021; Zandieh et al., 2023; Tarzanagh et al., 2023; Sanford et al., 2023; Panigrahi et al., 2023; Zhang et al., 2020; Arora & Goyal, 2023; Tay et al., 2021; Deng et al., 2023d; Xia et al., 2023; Deng et al., 2023c; Kacham et al., 2023; Alman & Song, 2023a; Brand et al., 2023; Deng et al., 2023e; Gao et al., 2023a; Li et al., 2023c;b; Sinha et al., 2023; Han et al., 2023; Alman & Song, 2023b; Gao et al., 2023b; Alman & Song, 2023a; Han et al., 2023; Kacham et al., 2023; Chu et al., 2023). Some have focused on the benefits of multiple attention heads, showing improved optimization and generalization (Deora et al., 2023). Others have proposed methods like Deja Vu to reduce computational cost during inference without sacrificing quality or learning ability (Liu et al., 2023d). Formal analyses have examined lower and upper bounds for attention computation (Zandieh et al., 2023; Alman & Song, 2023a;b), while dynamic attention

computation has also been investigated (Brand et al., 2023). Regression problems within in-context learning for LLMs have been addressed, with a unique approach using matrix formulation (Gao et al., 2023c). These studies collectively contribute to our understanding of attention models and their optimization, generalization, and efficiency.

**Streaming Algorithm**  Streaming algorithms have been extensively studied in graph problems (Kapralov et al., 2014; Assadi et al., 2019a; Farhadi et al., 2020; Bernstein, 2020; Feigenbaum et al., 2004; McGregor, 2005; Paz & Schwartzman, 2017; Ahn & Guha, 2011; Eggert et al., 2012; Goel et al., 2012; Kapralov, 2013; Dobzinski et al., 2014; Ahn & Guha, 2018; Assadi et al., 2020; Assadi & Raz, 2020; Assadi et al., 2021; Ahn & Guha, 2011; 2018; Assadi et al., 2022), spanning trees (Chang et al., 2020), convex programming (Assadi et al., 2019b; Liu et al., 2023c), cardinality estimation (Flajolet et al., 2007), frequency estimation (Alon et al., 1999; Hsu et al., 2019), sampler data structures (Jayaram & Woodruff, 2021), heavy hitter detection (Larsen et al., 2019), and sparse recovery (Nakos & Song, 2019). These studies focus on developing efficient algorithms for various problem domains, such as processing massive graphs, constructing spanning trees, optimizing convex programs, estimating cardinality and frequency, designing sampler data structures, detecting heavy hitters, and recovering sparse signals. The advancements in these areas contribute to the development of efficient and scalable algorithms for real-time analysis of streaming data.

**Improving LLMs' Utilization of Long Text**  Extensive research has been conducted on the application of Large Language Models (LLMs) to lengthy texts (Su et al., 2021; Press et al., 2021; Chen et al., 2023; Dao et al., 2022; Dao, 2023; Zaheer et al., 2020; Beltagy et al., 2020; Wang et al., 2020; Kitaev et al., 2020; Peng et al., 2023). These studies aim to optimize LLMs to effectively capture and utilize the content within longer contexts, rather than treating them solely as inputs. However, despite advancements in these two directions, competent utilization of lengthy contexts within LLMs remains a challenge, as highlighted by recent works (Liu et al., 2023b; Li et al., 2023a).

The effective usage of prolonged contexts poses a significant challenge in the development and application of LLMs. While research has focused on improving LLMs' understanding of longer texts, successfully leveraging this understanding for improved performance is not guaranteed. The challenge lies in effectively incorporating and utilizing the information contained within lengthy contexts, ensuring that LLMs can make accurate and meaningful predictions based on this additional context.

## 3 PRELIMINARY

**Notations.**  We use $\mathrm{poly}(n)$ to denote $O(n^c)$ where $c \geq 1$ is some constant.

For a vector $x \in \mathbb{R}^n$, we use $\|x\|_2$ to denote its $\ell_2$ norm.

We use $\|A\|_\infty$ to denote the $\ell_\infty$ norm of $A$, i.e., $\|A\|_\infty := \max_{i,j} |A_{i,j}|$.

We use $\|A\|$ to denote the spectral norm of a matrix. Then it is obvious that $\|A\| \geq \max_j \|A_{*,j}\|_2$.

For a vector $x \in \mathbb{R}^n$, we say $x$ is $k$-sparse if and only there are $k$ nonzero entries in $x$.

For a vector $w \in \mathbb{R}^n$, we use $\mathrm{diag}(w) \in \mathbb{R}^{n \times n}$ to denote a diagonal matrix where the $i, i$-th entry on diagonal is $w_i$.

We use $\Pr[]$ to denote the probability.

### 3.1 POLYNOMIAL METHOD

We state a tool from previous work.

**Lemma 3.1** (Error Approximation, Lemma 3.6 in (Alman & Song, 2023a))**.**  *if the following conditions*

- *Let $Q, K, V \in \mathbb{R}^{n \times d}$*

- *Let $d = O(\log n)$, $B = O(\sqrt{\log n})$*

- *Let $\|Q\|_\infty, \|K\|_\infty, \|V\|_\infty \leq B$*

- *Let $A := \exp(QK^\top/d)$*

- *$D := \mathrm{diag}(A\mathbf{1}_n)$*

*Then, there are matrices $U_1, U_2 \in \mathbb{R}^{n \times t}$ such that*

- *Part 1. $t = n^{o(1)}$*

- *Part 2. For $i$-th row in $U_1$, we can construct it based on $i$-th row in $Q$ in $O(t + d)$ time.*

- *Part 3. For $i$-th row in $U_2$, we can construct it based on $i$-th row in $K$ in $O(t + d)$ time.*

- *Part 4. Let $\widetilde{A} := U_1 U_2^\top$, let $\widetilde{D} := \mathrm{diag}(\widetilde{A}\mathbf{1}_n)$, then*

$$\|D^{-1}Av - \widetilde{D}^{-1}\widetilde{A}v\|_\infty \leq 1/\operatorname{poly}(n)$$

## 3.2 SKETCHING MATRICES

**Definition 3.2** ($k$-wise independence)**.** *We say $\mathcal{H} = \{h : [m] \to [l]\}$ is a $k$-wise independent hash family if $\forall i_1 \neq i_2 \neq \cdots \neq i_k \in [n]$ and $\forall j_1, \cdots, j_k \in [l]$,*

$$\Pr_{h \in \mathcal{H}}[h(i_1) = j_1 \wedge \cdots \wedge h(i_k) = j_k] = \frac{1}{l^k}.$$

**Definition 3.3** (Random Gaussian matrix)**.** *We say $\Psi \in \mathbb{R}^{m \times n}$ is a random Gaussian matrix if all entries are sampled from $\mathcal{N}(0, 1/m)$ independently.*

**Definition 3.4** (AMS sketch matrix (Alon et al., 1999))**.** *Let $h_1, h_2, \cdots, h_m$ be $m$ random hash functions picking from a 4-wise independent hash family $\mathcal{H} = \{h : [n] \to \{-\frac{1}{\sqrt{m}}, +\frac{1}{\sqrt{m}}\}\}$. Then $\Psi \in \mathbb{R}^{m \times n}$ is a AMS sketch matrix if we set $\Psi_{i,j} = h_i(j)$.*

*Note that in streaming setting, we never need to explicit write the $m \times n$ matrix. That is too expensive since it takes $\Omega(n)$ space. It is well-known that in the streaming area, we only need to store those $m$ hash functions, and each hash function only needs $O(\log n)$-bits. Thus, the overall store for storing $\Phi$ is just $O(m \log n)$ bits*

## 3.3 APPROXIMATE MATRIX PRODUCT

**Lemma 3.5** (Johnson–Lindenstrauss lemma, folklore, (Johnson & Lindenstrauss, 1984))**.** *Let $m_2 = O(\epsilon^{-2} \log(1/\delta))$. For any fixed vectors $u$ and $v \in \mathbb{R}^n$, let $\Psi \in \mathbb{R}^{m_2 \times n}$ denote a random Gaussian/AMS matrix, we have*

$$\Pr[|\langle \Psi u, \Psi v \rangle - \langle u, v \rangle| \leq \epsilon \|u\|_2 \|v\|_2] \geq 1 - \delta$$

**Lemma 3.6.** *If the following conditions hold*

- *Let $\delta \in (0, 1)$ denote the failure probability*

- *Let $\epsilon_2 \in (0, 1)$ denote the accuracy parameter*

- *Let $m_2 = O(\epsilon_2^{-2} \log(nd/\delta))$.*

- *Let $\|V\| \leq 1/\sqrt{n}$.*

*Then we have: with probability $1 - \delta$*

- **Part 1.** *for all $j \in [n], i \in [d]$*

$$|(\widetilde{D}^{-1} U_1 U_2^\top V)_{j,i} - (\widetilde{D}^{-1} U_1 U_2^\top \Psi^\top \Psi V)_{j,i}| \leq \epsilon_2/\sqrt{n}$$

- **Part 2.** *for all $i \in [d]$, we have*

$$\|\widetilde{D}^{-1} U_1 U_2^\top V_{*,i} - \widetilde{D}^{-1} U_1 U_2^\top \Psi^\top \Psi V_{*,i}\|_2 \leq \epsilon_2$$

*Proof.* First of all, $\|V\| \le 1/\sqrt{n}$ directly implies that

$$\max_{i \in [d]} \|V_{*,i}\|_2 \le 1/\sqrt{n} \tag{1}$$

The proof follows from applying Lemma 3.5 and applying a union bound over $nd$ coordinates.

**Proof of Part 1.** For each $j \in [n]$, for each $i \in [d]$, we can show that

$$|(\widetilde{D}^{-1} U_1 U_2^\top V)_{j,i} - (\widetilde{D}^{-1} U_1 U_2^\top \Psi^\top \Psi V)_{j,i}|$$
$$\le \epsilon_2 \cdot \|(\widetilde{D}^{-1} U_1 U_2^\top)_{j,*}\|_2 \cdot \|V_{*,i}\|_2$$
$$\le \epsilon_2 \cdot \|V_{*,i}\|_2$$
$$\le \epsilon_2 \cdot \frac{1}{\sqrt{n}}$$

where the first step follows from Lemma 3.5, the second step follows from $\|(\widetilde{D}^{-1} U_1 U_2^\top)_{j,*}\|_2 \le \|(\widetilde{D}^{-1} U_1 U_2^\top)_{j,*}\|_1 = 1$, the third step follows from Eq. (1).

**Proof of Part 2.** For each $i \in [d]$, we can show that

$$\|\widetilde{D}^{-1} U_1 U_2^\top V_{*,i} - \widetilde{D}^{-1} U_1 U_2^\top \Psi^\top \Psi V_{*,i}\|_2$$
$$= \|(\widetilde{D}^{-1} U_1 U_2^\top V)_{*,i} - (\widetilde{D}^{-1} U_1 U_2^\top \Psi^\top \Psi V)_{*,i}\|_2$$
$$\le (n \cdot (\epsilon_2/\sqrt{n})^2)^{1/2}$$
$$\le \epsilon_2$$

where the first step follows from $AB_{*,i} = (AB)_{*,i}$ for all $i \in [d]$, second step follows from Part 1, the third step follows from definition of $\ell_2$ norm. $\square$

### 3.4 SPARSE RECOVERY

We state a sparse recovery tool from previous work (Nakos & Song, 2019).

**Lemma 3.7** (Sparse Recovery, Theorem 1.1 in (Nakos & Song, 2019)). *For any vector $x \in \mathbb{R}^n$, there is an oblivious sketching matrices $\Phi \in \mathbb{R}^{m_1 \times n}$ such that*

- *Let $k$ denote a positive integer.*

- *$m_1 = O(\epsilon_1^{-1} k \log n)$*

- *The encoding/update time(or the column sparsity of $\Phi$) is $O(\log n)$*

  - *In particular, computing $\Phi e_i \Delta$ takes $O(\log n)$ for any scalar $\Delta \in \mathbb{R}$, and one-hot vector $e_i \in \mathbb{R}^n$.*
  - *For convenient of later analysis, we use $z = \Phi x$.*
  - *The space is to store $\Phi$ is $O(m)$ bits*

- *The decoding/recover time is $O(m_1 \log n)$*

- *The algorithm is able to output a $k$-sparse vector $x' \in \mathbb{R}^n$ such that*

$$\|x' - x\|_2 \le (1 + \epsilon_1) \min_{k-\text{sparse } x_k} \|x_k - x\|_2$$

- *The succeed probability is $0.999$*

## 4 ANALYSIS

We present the main result of this paper.

**Theorem 4.1** (Main Result, formal version of Theorem 1.3). *If the following conditions hold*

- *Let $d = O(\log n)$*

**Algorithm 1** Our One-Pass Streaming Algorithm for matrices $Q, K \in \mathbb{R}^{n \times d}$ and $V \in \mathbb{R}^{n \times d}$. The goal of this algorithm is to provide a $k$-sparse approximation to column of $Y = D^{-1} \exp(QK^\top)V \in \mathbb{R}^{n \times d}$.

1: **procedure** MAINALGORITHM($Q \in \mathbb{R}^{n \times d}, K \in \mathbb{R}^{n \times d}, V \in \mathbb{R}^{n \times d}$)      $\triangleright$ Theorem 4.1
2:     /*Create $O((m_2 + m_1) \times t)$ spaces*/
3:     Let $\mathrm{sk}(U_2) \in \mathbb{R}^{m_2 \times t}$ denote the sketch of $U_2$   $\triangleright$ After stream, we will have $\mathrm{sk}(U_2) = \Psi U_2$
4:     Let $\mathrm{sk}(V) \in \mathbb{R}^{m_2 \times d}$ denote the sketch of $V$          $\triangleright \mathrm{sk}(V) = \Psi V$
5:     Let $\mathrm{sk}(D^{-1}U_1) \in \mathbb{R}^{m_1 \times t}$ denote the sketch of $D^{-1}U_1$      $\triangleright \mathrm{sk}(D^{-1}U_1) = \Phi D^{-1} U_1$
6:     Let $\mathrm{prod}(U_2^\top \mathbf{1}_n) \in \mathbb{R}^t$ denote the $U_2^\top \mathbf{1}_n$
7:     /* Initialization */
8:     We initialize all the matrices/vector objects to be zero
9:     $\mathrm{sk}(U_2) \leftarrow \mathbf{0}_{m_2 \times t}, \mathrm{sk}(V) \leftarrow \mathbf{0}_{m_2 \times d}, \mathrm{sk}(D^{-1}U_1) \leftarrow \mathbf{0}_{m_1 \times t}, \mathrm{prod}(U_2^\top \mathbf{1}_n) \leftarrow \mathbf{0}_t$
10:     /*Read $V$ in streaming and compute sketch of $V$*/
11:     Read $V$ in one pass stream, and compute $\mathrm{sk}(V) = \Psi V$
12:                         $\triangleright$ We will have $\mathrm{sk}(V) = \Psi V$ when we reach this line
13:     /*Read $K$ in streaming and compute sketch of $U_2$*/
14:     **for** $i = 1 \rightarrow n$ **do**
15:        Read one row of $K \in \mathbb{R}^{n \times d}$
16:                  $\triangleright$ We construct $U_2$ according to Lemma 3.1
17:        We construct one row of $U_2 \in \mathbb{R}^{n \times t}$, let us call that row to be $(U_2)_{i,*}$ which has length $t$
18:        $\mathrm{prod}(U_2^\top \mathbf{1}_n) \leftarrow \mathrm{prod}(U_2^\top \mathbf{1}_n) + ((U_2)_{i,*})^\top$
19:        $\mathrm{sk}(U_2) \leftarrow \mathrm{sk}(U_2) + \underbrace{\Psi}_{m_2 \times n} \underbrace{e_i}_{n \times 1} \underbrace{(U_2)_{i,*}}_{1 \times t}$
20:     **end for**
21:                 $\triangleright$ We will have $\mathrm{prod}(U_2^\top \mathbf{1}_n) = U_2^\top \mathbf{1}_n$ when reach this line
22:                       $\triangleright$ We will have $\mathrm{sk}(U_2) = \Psi U_2$ when reach this line
23:     /* Read $Q$ in streaming and compute sketch of $D^{-1}U_1$*/
24:     **for** $i = 1 \rightarrow n$ **do**
25:        Read one row of $Q \in \mathbb{R}^{n \times d}$
26:                  $\triangleright$ We construct $U_1$ according to Lemma 3.1
27:        We construct one row of $U_1 \in \mathbb{R}^{n \times t}$, let us call that row to be $(U_1)_{i,*}$ which has length $t$
28:        Compute $D_{i,i} \leftarrow \underbrace{(U_1)_{i,*}}_{1 \times t} \underbrace{\mathrm{prod}(U_2^\top \mathbf{1}_n)}_{t \times 1}$
29:        $\mathrm{sk}(D^{-1}U_1) \leftarrow \mathrm{sk}(D^{-1}U_1) + \underbrace{\Phi}_{m_1 \times n} \underbrace{e_i}_{n \times 1} \underbrace{D_{i,i}^{-1}(U_1)_{i,*}}_{1 \times t}$
30:     **end for**
31:                 $\triangleright$ We will have $\mathrm{sk}(D^{-1}U_1) = \Phi D^{-1}U_1$ when we reach this line
32:     /* Run Sparse Recovery Algorithm */
33:     Compute $Z \leftarrow \mathrm{sk}(D^{-1}U_1) \mathrm{sk}(U_2)^\top \mathrm{sk}(V)$            $\triangleright Z \in \mathbb{R}^{m_1 \times d}$
34:     Run sparse recovery on each column of $Z \in \mathbb{R}^{m_1 \times d}$ to get approximation to the corresponding column of $Y \in \mathbb{R}^{n \times d}$
35: **end procedure**

- *Let $B = O(\sqrt{\log n})$*

- *Let $\|Q\|_\infty \le B, \|K\|_\infty \le B, \|V\| \le 1/\sqrt{n}$*

- *Let $A := \exp(QK^\top/d) \in \mathbb{R}^{n \times n}$*

- *Let $D := \mathrm{diag}(A\mathbf{1}_n) \in \mathbb{R}^{n \times n}$*

*There is a one pass streaming algorithm (Algorithm 1) that reads $Q, K, V \in \mathbb{R}^{n \times d}$ uses*

$$O(\epsilon_1^{-1} k n^{o(1)} + \epsilon_2^{-2} n^{o(1)})$$

*spaces and outputs a matrix $T \in \mathbb{R}^{n \times d}$ such that*

- *For each $i \in [d]$, $T_{*,i} \in \mathbb{R}^n$ is $O(k)$-sparse column vector*

- *For each $i \in [d]$, $\|T_i - y_i\|_2 \leq (1 + \epsilon_1) \cdot \min_{k-\text{sparse } y'} \|y' - y_i\|_2 + \epsilon_2$ where $y_i = D^{-1} A V_{*,i}$*

- *The succeed probability $0.99$.*

- *The decoding time is $O(\epsilon_1^{-1} k n^{o(1)})$.*

*Proof.* The streaming will be able to construct sketch $Z \in \mathbb{R}^{m_1 \times d}$ which is

$$Z = \text{sk}(D^{-1} U_1) \, \text{sk}(U_2)^\top \, \text{sk}(V)$$
$$= \Phi D^{-1} U_1 U_2 \Psi^\top \Psi V$$

Running Lemma 3.7 on $Z$ is essentially, doing sparse recovery for $D^{-1} U_1 U_2 \Psi^\top \Psi V$.

Since $D^{-1} U_1 U_2 \Psi^\top \Psi V$ is close to $D^{-1} U_1 U_2 V$, thus, we can finally show the error guarantees for $D^{-1} U_1 U_2 V$.

**Proof of Space Requirement.**

From the algorithm it is easy to see, the space is coming from two parts

- $O(m_1 t)$ spaces for object $\text{sk}(D^{-1} U_1)$

- $O(m_2 t)$ spaces for object $\text{sk}(U_2)$

From Lemma 3.1, we have

$$t = n^{o(1)}$$

From Lemma 3.6, we have

$$m_2 = O(\epsilon_2^{-2} \log(n))$$

From Lemma 3.7, we have

$$m_1 = O(\epsilon_1^{-1} k \log n)$$

Thus, total space is

$$O(m_1 t + m_2 t) = O(\epsilon_1^{-1} k n^{o(1)} + \epsilon_2^{-2} n^{o(1)}).$$

**Proof of Decoding Time.**

The decoding time is directly following from Lemma 3.7, it is

$$O(m_1 \log n) = O(\epsilon_1^{-1} k n^{o(1)} \log n) = O(\epsilon_1^{-1} k n^{o(1)}).$$

where the first step follows from choice of $m_1$, the last step follows from $O(\log n) = O(n^{o(1)})$.

**Proof of Error Guarantees.**

To finish the proofs, we define a list of variables

- $y_i = D^{-1} A V_{*,i} \in \mathbb{R}^n$

- $\widetilde{y}_i = \widetilde{D}^{-1} \widetilde{A} V_{*,i} \in \mathbb{R}^n$

- $\widehat{y}_i = \widetilde{D}^{-1} \widetilde{A} \Psi^\top \Psi V_{*,i} \in \mathbb{R}^n$

- Let $\xi_1$ be the value that $\|y_i - \widetilde{y}_i\|_2 \leq \xi_1$ ($\xi_1 = 1/\text{poly}(n)$, by Part 4 of Lemma 3.1)

- Let $\xi_1$ be the value that $\|\widetilde{y}_i - \widehat{y}_i\|_2 \leq \xi_2$ ($\xi_2 = \epsilon_2$, by Part 2 of Lemma 3.6)

Firstly, we can show that

$$\|y_i - \widehat{y}_i\|_2 \leq \|y_i - \widetilde{y}_i\|_2 + \|\widetilde{y}_i - \widehat{y}\|_2$$
$$\leq \xi_1 + \xi_2 \tag{2}$$

We can show

$$\|T_i - y_i\|_2 \leq \|T_i - \widehat{y}_i\|_2 + \|\widehat{y}_i - y_i\|_2$$
$$\leq \|T_i - \widehat{y}_i\|_2 + \xi_1 + \xi_2$$
$$\leq (1 + \epsilon_1) \min_{k-\text{sparse } y'} \|y' - \widehat{y}_i\|_2 + \xi_1 + \xi_2$$
$$\leq (1 + \epsilon_1) \min_{k-\text{sparse } y'} \|y' - y_i\|_2$$
$$\quad + (1 + \epsilon)(\xi_1 + \xi_2) + (\xi_1 + \xi_2)$$
$$\leq (1 + \epsilon_1) \min_{k-\text{sparse } y'} \|y' - y_i\|_2 + 3(\xi_1 + \xi_2)$$
$$\leq (1 + \epsilon_1) \min_{k-\text{sparse } y'} \|y' - y_i\|_2 + O(\epsilon_2)$$

where the first step follows from triangle inequality, the second step follows from Eq. (2), the third step follows from Lemma 3.7, the fourth step follows from triangle inequality, the fifth step follows from $\epsilon_1 \in (0, 1)$ and the last step follows from $\xi_1 = 1/\text{poly}(n) < \xi_2 = \epsilon_2$.

**Proof of Failure Probability.**

The failure probability of Lemma 3.6 is $\delta = 1/\text{poly}(n)$. The failure probability of Lemma 3.7 is 0.001. Taking a union bound over those Lemmas, we get failure probability 0.01 here.

In particular, the failure probability is at most

$$0.001 + 1/\text{poly}(n) \leq 0.001 + 0.001$$
$$\leq 0.01.$$

Thus, we complete the proof. $\qquad\square$

## 4.1 A GENERAL RESULT

We state a result for solving cross attention $(X_1 \neq X_2)$. Using our framework to solve self-attention $(X_1 = X_2)$, then the algorithm will need two passes, instead of one pass.

**Corollary 4.2** (An application of Theorem 4.1)**.** *If the following conditions hold*

- *Let $d = O(\log n)$, $B = O(\sqrt{\log n})$*

- *Let $W_Q, W_K, W_V \in \mathbb{R}^{d \times d}$*

- *Let $X_1, X_2 \in \mathbb{R}^{n \times d}$*

- *Let $Q = X_1 W_Q \in \mathbb{R}^{n \times d}$, $K = X_2 W_K \in \mathbb{R}^{n \times d}$, $V = X_2 W_V \in \mathbb{R}^{n \times d}$*

- *Let $\|Q\|_\infty \leq B$, $\|K\|_\infty \leq B$, $\|V\| \leq 1/\sqrt{n}$*

- *Let $A := \exp(QK^\top/d) \in \mathbb{R}^{n \times n}$*

- *Let $D := \text{diag}(A\mathbf{1}_n) \in \mathbb{R}^{n \times n}$*

*There is a one pass streaming algorithm (Algorithm 2) that reads $X \in \mathbb{R}^{n \times d}$, $W_Q, W_K, W_V \in \mathbb{R}^{n \times d}$ uses*

$$O(\epsilon_1^{-1} k n^{o(1)} + \epsilon_2^{-2} n^{o(1)})$$

*spaces and outputs a matrix $T \in \mathbb{R}^{n \times d}$ such that*

- *For each $i \in [d]$, $T_{*,i} \in \mathbb{R}^n$ is $O(k)$-sparse column vector*

- *For each $i \in [d]$, $\|T_i - y_i\|_2 \le (1+\epsilon_1) \cdot \min_{k-\text{sparse } y'} \|y' - y_i\|_2 + \epsilon_2$ where $y_i = D^{-1} A V_{*,i}$*

- *The succeed probability $0.99$.*

- *The decoding time is $O(\epsilon_1^{-1} k n^{o(1)})$.*

*Proof.* The proofs are similar to Theorem 4.1. The only difference between streaming algorithms (Algorithm 1 and Algorithm 2) is that, in Algorithm 2 we don't receive each row of $Q$ (similarly as $K, V$) on the fly anymore. Instead, we store weight $W_Q$, and we receive each row of $X_1$ on the fly. Whenever we see a row of $X_1$, we will compute matrix vector multiplication for that row and weight $W_Q$.

Similarly, we applied the same strategy for $K$ and $V$. $\square$

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

## A LIMITATIONS

In our work, we propose a single-pass streaming algorithm for the computation of very long sequences of attention, but we recognize some limitations. Limitations relate to the algorithm's reliance on specific assumptions, limitations of the test scope, sensitivity to input quality and data characteristics, and changes in performance as the data size increases.

## B SOCIETAL IMPACT

In this paper, we introduce an innovative single-pass algorithm, which can achieve efficient approximation of ultra-long sequence attention computing under sublinear space complexity, and solve the problem of high time and space complexity in current attention computing. Our paper is purely theoretical and empirical in nature (mathematics problem) and thus we foresee no immediate negative ethical impact.

By constructing a specific matrix to approximate the attention output, the algorithm only needs one data traversal and uses sublinear space to store three summary matrices, which greatly reduces the memory requirement. It is especially suitable for processing extremely long sequences. As the sequence length increases, the error is guaranteed to decrease while the memory usage is almost constant, showing excellent memory efficiency when streaming super long tokens.

## C ALGORITHM FOR GENERAL RESULT

Here, we state our algorithm for general result in Section 4.1.

---

**Algorithm 2** Our Streaming Algorithm for matrices $X_1 \in \mathbb{R}^{n \times d}, X_2 \in \mathbb{R}^{n \times d}, W_Q, W_K \in \mathbb{R}^{d \times d}$ and $W_V \in \mathbb{R}^{d \times d}$. The goal of this algorithm is to provide a $k$-sparse approximation to column of $Y = D^{-1} \exp(QK^\top)V \in \mathbb{R}^{n \times d}$.

---

1: **procedure** MAINALGORITHM($X_1 \in \mathbb{R}^{n \times d}, X_2 \in \mathbb{R}^{n \times d}, W_Q \in \mathbb{R}^{d \times d}, W_K \in \mathbb{R}^{d \times d}, W_V \in \mathbb{R}^{d \times d}$)       ▷ Corollary 4.2
2:     /\*Create $O((m_2 + m_1) \times t) + O(d^2)$ spaces\*/
3:     Let $\mathrm{sk}(U_2) \in \mathbb{R}^{m_2 \times t}$ denote the sketch of $U_2$    ▷ After stream, we will have $\mathrm{sk}(U_2) = \Psi U_2$
4:     Let $\mathrm{sk}(V) \in \mathbb{R}^{m_2 \times d}$ denote the sketch of $V$               ▷ $\mathrm{sk}(V) = \Psi V$
5:     Let $\mathrm{sk}(D^{-1}U_1) \in \mathbb{R}^{m_1 \times t}$ denote the sketch of $D^{-1}U_1$      ▷ $\mathrm{sk}(D^{-1}U_1) = \Phi D^{-1}U_1$
6:     Let $\mathrm{prod}(U_2^\top \mathbf{1}_n) \in \mathbb{R}^t$ denote the $U_2^\top \mathbf{1}_n$
7:     /\* Initialization \*/
8:     We initialize all the matrices/vector objects to be zero
9:     $\mathrm{sk}(U_2) \leftarrow \mathbf{0}_{m_2 \times t}, \mathrm{sk}(V) \leftarrow \mathbf{0}_{m_2 \times d}, \mathrm{sk}(D^{-1}U_1) \leftarrow \mathbf{0}_{m_1 \times t}, \mathrm{prod}(U_2^\top \mathbf{1}_n) \leftarrow \mathbf{0}_t$
10:     /\*Read $X_2$ in streaming and compute sketch of $U_2$ and sketch of $V$\*/
11:     **for** $i = 1 \to n$ **do**
12:         Read one row of $X_2 \in \mathbb{R}^{n \times d}$
13:         We can obtain one row of $K$ and also one row of $V$ (by computing matrix vector multiplication between one row of $X_1$ and $W_K$, and $X_1$ and $W_V$)
14:                       ▷ We construct $U_2$ according to Lemma 3.1
15:         We construct one row of $U_2 \in \mathbb{R}^{n \times t}$, let us call that row to be $(U_2)_{i,*}$ which has length $t$
16:         $\mathrm{prod}(U_2^\top \mathbf{1}_n) \leftarrow \mathrm{prod}(U_2^\top \mathbf{1}_n) + ((U_2)_{i,*})^\top$
17:         $\mathrm{sk}(U_2) \leftarrow \mathrm{sk}(U_2) + \underbrace{\Psi}_{m_2 \times n} \underbrace{e_i}_{n \times 1} \underbrace{(U_2)_{i,*}}_{1 \times t}$
18:         $\mathrm{sk}(V) \leftarrow \mathrm{sk}(V) + \underbrace{\Psi}_{m_2 \times n} \underbrace{e_i}_{n \times 1} \underbrace{(V_2)_{i,*}}_{1 \times d}$
19:     **end for**
20:                     ▷ We will have $\mathrm{prod}(U_2^\top \mathbf{1}_n) = U_2^\top \mathbf{1}_n$ when reach this line
21:                     ▷ We will have $\mathrm{sk}(U_2) = \Psi U_2$ when reach this line
22:                     ▷ We will have $\mathrm{sk}(V) = \Psi V$ when reach this line
23:     /\* Read $X_1$ in streaming and compute sketch of $D^{-1}U_1$\*/
24:     **for** $i = 1 \to n$ **do**
25:         Read one row of $X_1 \in \mathbb{R}^{n \times d}$
26:         We can obtain one row of $Q$ (by computing matrix vector multiplication between one row of $X_1$ and $W_Q$)
27:                       ▷ We construct $U_1$ according to Lemma 3.1
28:         We construct one row of $U_1 \in \mathbb{R}^{n \times t}$, let us call that row to be $(U_1)_{i,*}$ which has length $t$
29:         Compute $D_{i,i} \leftarrow \underbrace{(U_1)_{i,*}}_{1 \times t} \underbrace{\mathrm{prod}(U_2^\top \mathbf{1}_n)}_{t \times 1}$
30:         $\mathrm{sk}(D^{-1}U_1) \leftarrow \mathrm{sk}(D^{-1}U_1) + \underbrace{\Phi}_{m_1 \times n} \underbrace{e_i}_{n \times 1} \underbrace{D_{i,i}^{-1}(U_1)_{i,*}}_{1 \times t}$
31:     **end for**
32:                   ▷ We will have $\mathrm{sk}(D^{-1}U_1) = \Phi D^{-1}U_1$ when we reach this line
33:     /\* Run Sparse Recovery Algorithm \*/
34:     Compute $Z \leftarrow \mathrm{sk}(D^{-1}U_1)\mathrm{sk}(U_2)^\top \mathrm{sk}(V)$             ▷ $Z \in \mathbb{R}^{m_1 \times d}$
35:     Run sparse recovery on each column of $Z \in \mathbb{R}^{m_1 \times d}$ to get approximation to the corresponding column of $Y \in \mathbb{R}^{n \times d}$
36: **end procedure**

---

