# OpenReview forum: "One Pass Streaming Algorithm for Super Long Token Attention Approximation in Sublinear Space"
_ICLR.cc/2025/Conference — Submitted to ICLR 2025_

### Official Review · Reviewer_AW92 · 2024-10-16

**Soundness:** 1
**Presentation:** 1
**Contribution:** 1
**Rating:** 3
**Confidence:** 3

**Summary:**

The paper presents a one pass streaming algorithm for computing attention in sublinear space when d is logarithmic.

**Strengths:**

Addressing LLMs with very large context length appears important and timely.

**Weaknesses:**

- The writing is poor, the paper is littered with typos and the general structure is very hard to follow. I found the paper very hard to read. Other comments aside, it cannot be published in its current state.
- I find it hard to evaluate the novelty of this paper. The related work section seems to be just a lump of citations, and it is not clear what is the actual contribution. It looks like the authors simply joined existing techniques to get the results.
- I am not sure about the motivation for a streaming computation. In what scenario will the computation be executed in a streaming fashion (i.e., the matrices need to be stored somewhere, right?)
- There is no experimental evaluation.

**Questions:**

See weaknesses.

---

### Official Review · Reviewer_1DSk · 2024-11-03

**Soundness:** 2
**Presentation:** 2
**Contribution:** 1
**Rating:** 1
**Confidence:** 5

**Summary:**

In this work, the authors study improving the time and space complexity of the attention mechanism in the non-causal setting. They give a single pass streaming algorithm that given the query, key and value vectors approximates the attention output using tools from randomized linear algebra and compressed sensing. At a high level, the algorithm computes sketches of low rank matrices $U_1, U_2 \in \mathbb{R}^{n \times k}$ so that sparse approximations for the columns of the attention output $\exp(QK^T / d)V$ can be computed using the sketches of the low rank matrices. The space required by the algorithm is $n^{o(1)}$ and the time to process the streams is $n^{1+o(1)}$.

**Strengths:**

--

**Weaknesses:**

1. Attention computation can be done without using $O(n^2)$ space. See Rabe and Staats "Self-Attention does not need $O(n^2)$ memory".
2. The version of the problem being studied i.e., without using causal masking is not super relevant to the current LLMs which all use causal masking.
3. As stated, the results only output at most $k \cdot d$ nonzero entries for the entire matrix $V$ which has $n$ rows. Since we expect each of the rows in the full attention output to have similar norms, many useful rows are completely marked to zero by this algorithm unless $k \ge n/d$ at which point the sketches stored by the algorithm are linear in size, at which point the claim of streaming algorithm does not make sense and the algorithm is essentially no different from Alman and Song.
4. No experimental verification of the ideas. At least showing how the algorithm works on small instances, even without implementing the additional sparse recovery steps would have been useful.

**Questions:**

--

---

### Official Review · Reviewer_sARr · 2024-11-08

**Soundness:** 3
**Presentation:** 3
**Contribution:** 1
**Rating:** 3
**Confidence:** 3

**Summary:**

The paper proposes a new framework for doing one pass streaming algorithm instead of the quadratic attention. This is useful when the sequence length $n$ is much larger than the dimension $d$ (i.e. $n\geq 2^d$).  The method relies heavily on the polynomial method proposed by Alman and Song which shows how we can compute matrices $U_1,U_2 $ such that $D^{-1} U_1^TU_2$ is approximately equal to the full attention function. They effectively employ a strategy whereby they sketch matrix U and then using sparse recovery tools to recover the original vector. Since this can be done in a  streaming fashion, one can get a substantially smaller algorithm with very few caveats.

**Strengths:**

The authors address a fundamental problem that is important. They produce an algorithm with clear guarantees. Furthermore most of the assumptions are clearly stated, the proofs easy to follow and the pseudocode is clean and easy to understand.

**Weaknesses:**

The author doesn't introduce any new techniques but this seems to be the application of the sketching machinery on top of the algorithm of Alman and Song.

The resulting algorithm has small asymptotic complexity but seems to be in extremely impractical in several ways. First it relies on $n$ being large $n \geq 2^d $ where as in practice $d \sim 10^4$ for even small transformers. As a result this assumption is quite strong. Secondly , the type of operations necessary are not easy to implement fast in modern hardware.

**Questions:**

N/A

---

### Meta-Review · Area_Chair_D6ut · 2024-12-15

**Metareview:**

The paper proposes a streaming algorithm for Attention computation. As mentioned by the reviewers the novelty is low and the paper cannot be accepted at this point.

**Additional Comments On Reviewer Discussion:**

The rebuttal does not address the concerns.

---

### Decision · Program_Chairs · 2025-01-22

Reject